# Pharmacokinetic Imaging Using ^99m^Tc-Mebrofenin to Untangle the Pattern of Hepatocyte Transporter Disruptions Induced by Endotoxemia in Rats

**DOI:** 10.3390/ph15040392

**Published:** 2022-03-24

**Authors:** Solène Marie, Irene Hernández-Lozano, Marc Le Vée, Louise Breuil, Wadad Saba, Maud Goislard, Sébastien Goutal, Charles Truillet, Oliver Langer, Olivier Fardel, Nicolas Tournier

**Affiliations:** 1Université Paris-Saclay, CEA, CNRS, Inserm, Laboratoire d’Imagerie Biomédicale Multimodale, BIOMAPS, Service Hospitalier Frédéric Joliot, 4 Place du Général Leclerc, 91401 Orsay, France; solene.marie@aphp.fr (S.M.); louise.breuil@universite-paris-saclay.fr (L.B.); wadad.saba@cea.fr (W.S.); maud.goislard@universite-paris-saclay.fr (M.G.); sebastien.goutal@universite-paris-saclay.fr (S.G.); charles.truillet@cea.fr (C.T.); 2Faculté de Pharmacie, Université Paris-Saclay, 92296 Châtenay-Malabry, France; 3AP-HP, Université Paris-Saclay, Hôpital Bicêtre, Pharmacie Clinique, 94270 Le Kremlin Bicêtre, France; 4Department of Clinical Pharmacology, Medical University of Vienna, 1090 Vienna, Austria; irene.hernandezlozano@meduniwien.ac.at (I.H.-L.); oliver.langer@meduniwien.ac.at (O.L.); 5Univ. Rennes, Inserm, EHESP, Irset (Institut de Recherche en Santé, Environnement et Travail)-UMR_S 1085, 35043 Rennes, France; marc.levee@univ-rennes1.fr; 6Univ. Rennes, CHU Rennes, Inserm, EHESP, Irset (Institut de Recherche en Santé, Environnement et Travail)-UMR_S 1085, 35043 Rennes, France; olivier.fardel@univ-rennes1.fr

**Keywords:** ABC-transporter, drug-induced liver injury, hepatotoxicity, organic anion-transporting polypeptide, pharmacokinetics, liver function, SLC-transporter

## Abstract

Endotoxemia-induced inflammation may impact the activity of hepatocyte transporters, which control the hepatobiliary elimination of drugs and bile acids. ^99m^Tc-mebrofenin is a non-metabolized substrate of transporters expressed at the different poles of hepatocytes. ^99m^Tc-mebrofenin imaging was performed in rats after the injection of lipopolysaccharide (LPS). Changes in transporter expression were assessed using quantitative polymerase chain reaction of resected liver samples. Moreover, the particular impact of pharmacokinetic drug–drug interactions in the context of endotoxemia was investigated using rifampicin (40 mg/kg), a potent inhibitor of hepatocyte transporters. LPS increased ^99m^Tc-mebrofenin exposure in the liver (1.7 ± 0.4-fold). Kinetic modeling revealed that endotoxemia did not impact the blood-to-liver uptake of ^99m^Tc-mebrofenin, which is mediated by organic anion-transporting polypeptide (Oatp) transporters. However, liver-to-bile and liver-to-blood efflux rates were dramatically decreased, leading to liver accumulation. The transcriptomic profile of hepatocyte transporters consistently showed a downregulation of multidrug resistance-associated proteins 2 and 3 (Mrp2 and Mrp3), which mediate the canalicular and sinusoidal efflux of ^99m^Tc-mebrofenin in hepatocytes, respectively. Rifampicin effectively blocked both the Oatp-mediated influx and the Mrp2/3-related efflux of ^99m^Tc-mebrofenin. The additive impact of endotoxemia and rifampicin led to a 3.0 ± 1.3-fold increase in blood exposure compared with healthy non-treated animals. ^99m^Tc-mebrofenin imaging is useful to investigate disease-associated change in hepatocyte transporter function.

## 1. Introduction

Inflammation is a common feature of many pathophysiological states, including infection, systemic diseases, and cancer [1]. The physiological changes accompanying inflammation may alter liver function, with consequences for the hepatobiliary elimination of endogenous and exogeneous compounds and lead to liver injury [2,3] or impact the pharmacokinetics (PK) of medications [4].

From a molecular perspective, liver function can be estimated by the activity of the membrane transporters expressed in hepatocytes, which work together to mediate the hepatobiliary elimination of bile salts, drugs, and metabolites [5]. Several solute carrier (SLC) influx transporters, including the organic anion-transporting polypeptides (OATP)1B1/3 (*SLCO1B1/1B3* corresponding to Oatp4/*Slco1b2* in rodents [6]), are expressed in the sinusoidal (blood-facing) membrane of hepatocytes, where they mediate the uptake of their substrates from blood into the liver (Figure 1) [7]. Adenosine triphosphate-binding cassette (ABC) efflux transporters expressed in the canalicular (bile-facing) membrane of hepatocytes, such as the human multidrug resistance-associated protein (MRP)2/*ABCC2* (Mrp2/*Abcc2* in rodents) or the P-glycoprotein (P-gp, encoded by the multidrug resistance 1 gene (*MDR1*/*ABCB1*) in humans and both *Mdr1a*/*Abcb1a* and *Mdr1b*/*Abcb1b* in rodents), control the biliary excretion of solutes and bile acids (Figure 1). Other ABC transporters such as human MRP3/*ABCC3* (Mrp3/*Abcc3* gene in rodents) mediate the sinusoidal efflux of drugs and metabolites from hepatocytes into the blood (Figure 1) [7].

Disruption of hepatocyte transporter function by inflammation is increasingly considered as a cause of disease-related changes in liver function, hepatobiliary elimination of drugs, PK, and toxicity [3,8]. Dysregulation of hepatocyte transporters and cytochromes has been reported during endotoxin-mediated inflammation induced by lipopolysaccharide (LPS) in vitro [9] and ex vivo, in several animal species [10,11,12,13]. In animal models, the impaired activity of ABC transporters during inflammation has been linked to cholestasis and/or abnormal liver accumulation of drugs and metabolites, which may provide a mechanistical explanation for drug-induced liver injury (DILI) [14]. A similar disruption in hepatocyte transporter function is likely to occur in patients and may lead to severe hepatotoxicity outcomes. It remains, however, technically difficult to untangle and estimate the intrinsic importance of each individual transporter in vivo, apart from cytochrome activity, in mediating inflammation-related changes in PK in animal models and patients [4].

Hepatocyte transporters are targets for pharmacokinetic drug–drug interactions (DDI) [15]. The antituberculosis drug rifampicin (rifampin) is known to inhibit the function of several hepatocyte transporters in vitro, including human OATPs, MRP2, and MRP3, and rat Oatp2/*Slco1a4*, Oatp4, and Mrp3 [16,17,18,19]. Single dose rifampicin (600 mg) was shown to enhance plasma exposure to OATP substrates [20], and is therefore recommended by regulatory agencies as a model inhibitor drug to conduct pharmacokinetic DDI studies in animals and humans [15]. However, the in vivo impact of rifampicin on each individual transporter function and the consequences of liver exposure to solutes cannot be assessed from the plasma PK. This particularly holds for transporters expressed at the canalicular pole of hepatocytes distant from the plasma compartment [21]. Moreover, most non-clinical/clinical PK and DDI studies are conducted in healthy subjects. This may not recapitulate the pathological context, and underestimates the impact of DDI in patients with inflammatory diseases [22]. 

^99m^Tc-mebrofenin is a radiopharmaceutical used for hepatobiliary scintigraphy, routinely used in hepatobiliary disorders to perform imaging of the liver, bile, and gallbladder to assess hepatic function and/or biliary excretion [23]. ^99m^Tc-mebrofenin is well characterized as a non-metabolized substrate of human OATP1B1, OATP1B3, MRP2, and MRP3, which govern its hepatobiliary clearance [23]. Preclinical studies conducted in Oatp- and Mrp2-deficient animals confirmed the transport of ^99m^Tc-mebrofenin by the rodent orthologs of these transporters [24,25]. Our team and other teams have shown that dynamic ^99m^Tc-mebrofenin imaging, aided by kinetic modeling, can be used for molecular imaging of hepatocyte transporter function at the different poles of hepatocytes in animals and humans [23,26,27,28] (Figure 1). Moreover, ^99m^Tc-mebrofenin allows for selectively and simultaneously assessing the impact of transporter-mediated DDIs at the sinusoidal or the canalicular level in vivo [29].

In this study, the impact of LPS-induced inflammation on liver transporter expression and function was investigated in rats, using quantitative transcriptomics and ^99m^Tc-mebrofenin imaging. Moreover, we hypothesized that inflammation may exacerbate the impact of transporter-mediated DDIs precipitated by rifampicin in terms of PK and/or liver exposure.

## 2. Results

### 2.1. Quantitative Transcriptomics

A significant decrease in the mRNA expression of several transporters was observed after LPS-treatment, including Mrp2 and Mrp3 (Figure 2 and Appendix A). Among the tested ABC transporters, the most important decrease was observed for the genes encoding P-gp, with a significant −68% decrease for *Mdr1a* and a −75% decrease for *Mdr1b*. The mRNA expression of most hepatic SLC transporters was also significantly decreased after LPS-treatment, except for organic anion transporter (Oat)2/*Slc22a7*. The decrease in Oatp1/*Slco1a1* mRNA expression was less pronounced than for Oatp2/*Slco1a4*. The decrease in Oatp4/*Slco1b2* expression was also not statistically significant. Moreover, qPCR showed a significant decrease in mRNA expression coding for cytochromes (Cyp), with a −74% decrease of *Cyp3a1* mRNA expression (Figure 2).

### 2.2. ^99m^Tc-Mebrofenin Imaging

Figure 3 shows the mean time activity curves (TACs) obtained in each studied group. The mean area under the curve (AUC) for each TAC is reported in Figure 4 and Appendix A. Compared with the healthy group, LPS significantly increased exposure to ^99m^Tc-mebrofenin in the liver (1.7 ± 0.4-fold, *p* < 0.01). A significant 1.5 ± 0.3-fold decrease of radioactivity in the intestine was also observed, suggesting reduced biliary excretion (*p* < 0.01).

High dose rifampicin (40 mg/kg) significantly reduced the amount of ^99m^Tc-mebrofenin in the intestines of healthy animals (1.9 ± 0.4-fold, *p* < 0.001) and LPS-treated animals (1.2 ± 0.3-fold, *p* < 0.001). High dose rifampicin did not impact liver exposure neither in healthy nor in LPS-treated animals. Rifampicin (9 or 40 mg/kg) increased blood exposure in healthy animals (unpaired *t*-test, *p* < 0.01), although no significant difference was found using a one-way ANOVA (*p* > 0.05, Figure 4). Rifampicin did not significantly increase blood exposure in LPS-treated animals, regardless of statistical analysis (*p* > 0.05, Figure 4). Impact of endotoxemic inflammation and rifampicin-perpetrated DDI on overall blood clearance were additive: blood exposure of ^99m^Tc-mebrofenin in LPS/rifampicin-treated animals was 3.0 ± 1.3-fold higher than for healthy non-treated animals (*p* < 0.01).

A pharmacological dose of rifampicin (9 mg/kg) significantly reduced the amount of ^99m^Tc-mebrofenin in the intestines in healthy animals (*p* < 0.001), reaching similar levels as for the high dose rifampicin-treated animals. The pharmacological dose of rifampicin did not further decrease intestinal radioactivity in LPS-treated animals (*p* > 0.05). The pharmacological dose of rifampicin did not impact liver exposure in healthy or LPS-treated animals (*p* > 0.05). The pharmacological dose of rifampicin did not significantly exacerbate the impact of LPS on blood exposure (*p* > 0.05, Figure 4).

Visually, the implemented PK model provided good fits for both the observed liver and intestinal radioactivity (Appendix A), and the parameter precision (assessed by percentage coefficient of variation, %CV) was, in general, acceptable (with a %CV of less than 40% for most of the subjects, Appendix A). A higher %CV was observed for four subjects in the estimation of *k*_2_ and *k*_3_, especially in the situation of complete inhibition. This probably reflects the difficulty to accurately estimate the extremely low values of the transfer rate constants (Appendix A). These subjects were not excluded from the statistical analysis. The rate constant *k*_1_ defining the radiotracer transfer from blood into liver was not significantly different between healthy and LPS-treated rats (Figure 5). Rifampicin dose-dependently decreased *k*_1_ to a similar extent for both healthy and LPS-treated animals (Figure 5). The sinusoidal efflux rate constant *k*_2_ was significantly lower in the LPS-treated animals compared with the healthy controls. Pharmacological dose of rifampicin significantly decreased *k*_2_ in healthy animals but caused no further *k*_2_ decrease in LPS-treated rats. A maximal decrease in *k*_2_ was observed after treatment with a high dose of rifampicin, in both healthy and LPS-treated animals. LPS exposure strikingly decreased the liver-to-bile transfer rate constant (*k*_3_) to a minimal level, which was not further decreased by rifampicin (Figure 5). The non-parametric test showed no significant differences in the values of *k*_5_, which estimates the rate transfer between the bile ducts and intestine, except between healthy and LPS-treated animals when treated with a pharmacological dose of rifampicin (Figure 5). However, the parameter precision was not satisfying for some subjects (Appendix A) and a high intra-group variability was observed (Figure 5).

## 3. Discussion

The present study highlights the dramatic impact of LPS-induced inflammation on hepatocyte transporter expression and function in rats. The consequences for carrier-mediated hepatobiliary clearance and for the magnitude of transporter-mediated DDIs were assessed in vivo using ^99m^Tc-mebrofenin imaging and kinetic modeling. 

Quantitative transcriptomics confirmed the important disruption of the expression of several hepatocyte transporters 24 h after LPS exposure. This is consistent with previous rat studies, in which the mRNA and protein levels of most hepatic transporters were dramatically decreased 6–12 h after LPS administration, although an increase in Mrp3 expression has been reported [10,11,12,30]. The concomitant decrease in the expression of cytochrome P450 justified the use of a metabolically-stable imaging probe for quantitative and selective determination of the functional impact of this downregulation on transporter expression [31]. The mechanisms for the dysregulation of transporters and cytochromes by LPS are thought to be mediated by xenobiotic receptors, such as pregnane X receptor (PXR), aryl hydrocarbon receptor (AhR), glucocorticoid receptor (GR), and constitutive androstane receptor (CAR) [32].

From a functional perspective, our results point to significant changes in the kinetics of ^99m^Tc-mebrofenin in LPS-treated rats, with a pronounced increase in liver exposure and a decreased bile content (Figure 4). The mechanistic importance of each individual transporter system was estimated using compartmental modeling. LPS significantly decreased both the canalicular (*k*_3_) and basolateral (*k*_2_) efflux rate of ^99m^Tc-mebrofenin, which may reflect the activity of MRP2 and MRP3, respectively (Figure 5). This is consistent with the observed decrease in the *Mrp2* and *Mrp3* expression (Figure 2). These changes led to a greater exposure to ^99m^Tc-mebrofenin in the liver of LPS-treated rats compared with healthy rats. Interestingly, LPS did not impact the Oatp-mediated uptake of ^99m^Tc-mebrofenin (estimated by *k*_1_), consistent with the limited and non-significant impact of LPS treatment on Oatp4 mRNA expression (Figure 2). 

The impact of two different doses of the OATP/MRP inhibitor rifampicin on hepatocyte transporter function was also assessed using ^99m^Tc-mebrofenin. In healthy rats, high dose rifampicin (40 mg/kg) almost completely inhibited the activity of Oatp, Mrp2, and Mrp3 inferred from the reduction in *k*_1_, *k*_3_, and *k*_2_, respectively, compared to the healthy untreated group (Figure 5). In rats, pharmacological dose rifampicin (9 mg/kg) did not significantly decrease Oatp function (although a tendency towards a decrease was observed), while basolateral Mrp3 and canalicular Mrp2 were almost entirely inhibited at this dose. This suggests differences in vulnerability to the inhibition of Oatp and Mrp2/3 at the different poles of hepatocytes by a same dose of rifampicin in vivo. This suggests that liver excretion of MRP2/3 substrates may be hindered, while uptake transport remains active during concomitant rifampicin therapy. This may lead to liver accumulation of concomitant drugs and bile acids, and account for the rifampicin-associated DILI observed in clinical practice [33,34].

The rifampicin-challenge was also tested in LPS-treated rats to investigate the magnitude of transporter-mediated DDIs during endotoxemia. In the presence of high-dose rifampicin (40 mg/kg), transfer rate constants were similar in healthy and LPS-treated animals (Figure 5). This may be explained by the almost complete inhibition of investigated transporters, achieved with the administration of high dose rifampicin, which was not further modulated by their LPS-mediated repression (Figure 5). However, the blood exposure to ^99m^Tc-mebrofenin in the situation of complete transporter inhibition (40 mg/kg rifampicin) was significantly higher in LPS-treated compared with healthy controls. This suggests a potentiation of the rifampicin perpetrated DDI in the context of endotoxemia, in an additive manner (Figure 4). This may be explained by the multi-organ dysfunction induced by endotoxemia, which may also impact the non-hepatic clearance of ^99m^Tc-mebrofenin, a phenomenon that is revealed when the carrier-mediated hepatobiliary route is blocked. Interestingly, the absence of an additional impact of high-dose rifampicin on Mrp2 and Mrp3 function in LPS-treated rats suggests that almost complete suppression of the transporter activity is obtained during endotoxemia. This supports a transporter-based mechanism for cholestasis observed during endotoxemia in animals and patients [10,35,36]. Pharmacological dose rifampicin did not exacerbate the impact of LPS on hepatocyte Mrp2 and Mrp3 function, which were already almost completely suppressed by LPS. Partial inhibition of the blood-to-liver transfer of ^99m^Tc-mebrofenin by a pharmacological dose of rifampicin was not exacerbated in LPS-treated animals, which further supports the negligible impact of endotoxemia on Oatp function (Figure 5).

In the absence of adequate methods, studies to address the impact of systemic inflammation or sepsis on PK are still rare and are mainly focused on metabolic enzyme activity [3,37]. ^99m^Tc-mebrofenin imaging in combination with kinetic modeling offers an appealing method to be safely translated into a clinical/hospital set-up to non-invasively investigate the impact of inflammation on the function of important hepatocyte transporters. ^9m^Tc-mebrofenin imaging may thus help guide precision medicine with better dose-selection for the numerous drugs, whose elimination depends on these particular transporters. This work illustrates the great translational potential of molecular imaging techniques to untangle the impact of drug transporters in controlling blood and liver exposure in healthy volunteers and patients [31].

## 4. Materials and Methods

### 4.1. Chemicals and Radiochemicals

LPS (from *Escherichia coli* serotype 0111:B4), was purchased from Sigma-Aldrich (Saint Quentin Fallavier, France) and rifampicin was used as the commercial drug Rifadine (Sanofi, Paris, France). Commercial kits of mebrofenin (Cholediam) were gifted from Mediam (Marcq en Baroeul, France). Each kit was labeled with a sodium ^99m^Tc-pertechnetate (^99m^Tc-TcO_4_Na) eluate (~750 MBq/mL) obtained from a sterile ^99^Mo/^99m^Tc generator (Tekcis, GE Healthcare, Vélizy-Villacoublay, France), followed by quality control according to the manufacturer’s recommendations.

### 4.2. Animals

A total of 41 male rats (Wistar, weight = 261 ± 108 g) aged of 5–7 weeks were used for this study. All animal experiments were in accordance with the recommendations of the European Community for animal experiments (2010/63/UE) and the French National Committees (law 2013-118) for the care and use of laboratory animals. The experimental protocol was approved by a local ethics committee for animal use (CETEA) and by the French ministry of agriculture (APAFIS#5375-20l60513 17426342, and 13 December 2018). Animals were housed in a controlled environment with access to food and water ad libitum. Half of the animals received an intraperitoneal injection of an extemporaneously prepared solution of LPS in physiological saline (2 mg/mL) at a dose of 4 mg/kg body weight. The mean weight loss observed 24 h after treatment by LPS was −18.8 ± 11.6% of the initial weight.

### 4.3. Transcriptomics in LPS-Treated Rats

The expression of the transcript of selected membrane transporters and metabolic enzymes was determined using quantitative polymerase chain reaction (qPCR) analysis, as previously described [38]. Twenty-four hours after LPS injection, eight animals were sacrificed by injecting pentobarbital, and the livers were excised and frozen in liquid nitrogen. The total RNAs were extracted from frozen liver fragments using the Nucleospin RNA kit (Macherey-Nagel, Düren, Germany) and the TissueLyser LT (Qiagen, Courtaboeuf, France) at 50 Hz for 2 min after RNA quantification using the NanoDrop ND-1000 Spectrophotomer (Thermo Fisher Scientific, Illkirch-Graffenstaden, France). RNAs were reversed transcribed using the Applied Biosystems cDNA Reverse Transcription kit (Thermo Fisher Scientific, Illkirch-Graffenstaden, France). Real-time quantitative PCR was performed using the fluorescent SYBR Green dye (Thermo Fisher Scientific, Illkirch-Graffenstaden, France) and the CFX384 TouchTM Real-time PCR Detection System (Bio-Rad, Marnes la Coquette, France). The primer sequences used for qPCR are detailed in Appendix A. Two independent liver fragments were processed and analyzed for each animal, and each measurement was performed in duplicate. The specificity of each gene amplification was verified at the end of quantitative PCR reactions, through analysis of the dissociation curves of the PCR products. Amplification curves were analyzed with CFX Manager Software (Bio-Rad, Marnes la Coquette, France), using the comparative cycle threshold method. Relative quantification of the steady-state target mRNA levels was calculated after normalization of the total amount of cDNA tested to the 18S rRNA endogenous reference, using the 2^(−ΔΔCt)^ method. Data were expressed as arbitrary units (a.u.) relative to the 18S rRNA contents, as previously reported [38]. The percentage change in mRNA expression in each individual LPS-treated rat (*n* = 4) was calculated compared with the mean gene expression obtained in four healthy rats.

### 4.4. ^99m^Tc-Mebrofenin Imaging

Imaging was performed in LPS-treated animals (24 h after LPS injection) and healthy animals (no LPS). In both groups, ^99m^Tc-mebrofenin scintigraphy was performed either under the baseline condition (no transporter inhibition) or after transporter inhibition using rifampicin. Two different doses of rifampicin were tested to investigate the different levels of transporter inhibition: 9 mg/kg (corresponding to a human pharmacological dose ~600 mg/70 kg) or 40 mg/kg (assuming maximal transporter inhibition at this dose) [29]. The number of investigated animals under each tested condition is reported in Appendix A.

Rats were anesthetized with isoflurane (3.5% and 1.5–2% in oxygen for induction and maintenance, respectively). ^99m^Tc-mebrofenin scintigraphy was performed using a clinical SPECT-CT camera (Symbia, Siemens, Knoxville, TN, USA) with a Low Energy High Resolution (LEHR) collimator. In each session, three rats were placed in a row on the scanner bed and catheters were inserted in the caudal vein. Scintigraphy imaging is not an absolute quantitative technique [31]. The position of the detectors relative to the scanner bed was standardized to limit the variability associated with in counting efficiency. Rifampicin was injected intravenously (i.v.), immediately (<5 s) before ^99m^Tc-mebrofenin injection. Dynamic planar scintigraphy acquisitions started with ^99m^Tc-mebrofenin injection (37.4 ± 5.0 MBq, i.v.) for 40 min, followed by an X-ray CT-scan. Dynamic images were reconstructed in 54 frames with time durations of 20 × 0.25 min, 10 × 0.5 min, 20 × 1 min, and 5 × 2 min.

### 4.5. Imaging Data Analysis

Images were analyzed with PMOD software (version 3.9, PMOD Technologies LLC, Zurich, Switzerland), as described in a previous rat study [29]. Regions of interest (ROI) were drawn on planar images over the liver and intestine (assumed to represent excreted bile). Standardized ROIs consisted in the largest part possible of each organ, excluding the overlapping region. The whole-heart was delineated to derive an image-derived blood input function, as previously described [29]. Corresponding TACs were generated by plotting the mean radioactivity counts (counts per second (cps)) in each region of interest normalized to the injected radioactivity amount in each animal (cps/MBq) versus time. 

A previously developed four-compartment pharmacokinetic model, which had already been applied to describe the transporter-mediated hepatobiliary disposition of ^99m^Tc-mebrofenin in rats [29,39], was used to estimate the rate constants that describe the transfer of the radiotracer between the blood and hepatocytes (*k*_1_ and *k*_2_, min^−1^), from hepatocytes to the intrahepatic bile ducts (*k*_3_, min^−1^), and from the intrahepatic bile ducts to bile excreted into the intestine (*k*_5_, min^−1^). The blood concentration was estimated from the total amount of radioactivity in the heart divided by a standard heart volume (V_heart_ = 2.55 mL), which was measured on an X-ray CT scan in a previous study in rats [29]. The model assumes that all radioactivity present in the intestine corresponds to excreted bile. In addition, the model includes a dual blood input function, which accounts for the radiotracer delivery to the liver via the hepatic artery and the portal vein. The hepatic artery concentration was obtained from the image-derived input function, while the concentration in the portal vein was mathematically estimated from the arterial blood curve and the liver and intestine scintigraphy data during the modeling process, as described previously [39]. The final flow-weighted dual-input blood curve was generated using a hepatic arterial flow fraction of 0.17 [29].

### 4.6. Statistical Analysis

Statistical analysis was performed using Prism 8 (GraphPad Software method, La Jolla, CA, USA) and *R* package (http://www.R-project.org/, accessed on 3 March 2022). Transcriptomic profiles were compared using a one-sample *t*-test after confirmation of normal distribution by a Shapiro–Wilk Normality test. Differences in pharmacokinetic parameters between study groups were assessed by ordinary one-way ANOVA, followed by a Tukey’s post hoc test for multiple comparison. Homoscedasticity was checked using the Levene’s test. Homoscedasticity was not confirmed for *k*_1_ and *k*_5_, for which a non-parametric Kruskal–Wallis test was performed. The level of statistical significance was set to a *p*-value < 0.05. Data are reported as mean ± standard deviation (S.D.).

## 5. Conclusions

^99m^Tc-mebrofenin imaging in rats unveiled a pattern of dramatic disruptions of hepatocyte transporters during LPS-induced endotoxemia in rats, particularly for Mrp2 and Mrp3 which control the biliary excretion and sinusoidal efflux of their substrates, respectively. This led to decreased hepatobiliary clearance and increased liver accumulation. High dose rifampicin almost completely blocked the carrier-mediated hepatobiliary transport of ^99m^Tc-mebrofenin, respectively. However, the impact of endotoxemia and transporter-mediated DDI induced by a high dose rifampicin were additive, suggesting the importance of multi-organ dysfunction in controlling blood exposure. 

## Figures and Tables

**Figure 1 pharmaceuticals-15-00392-f001:**
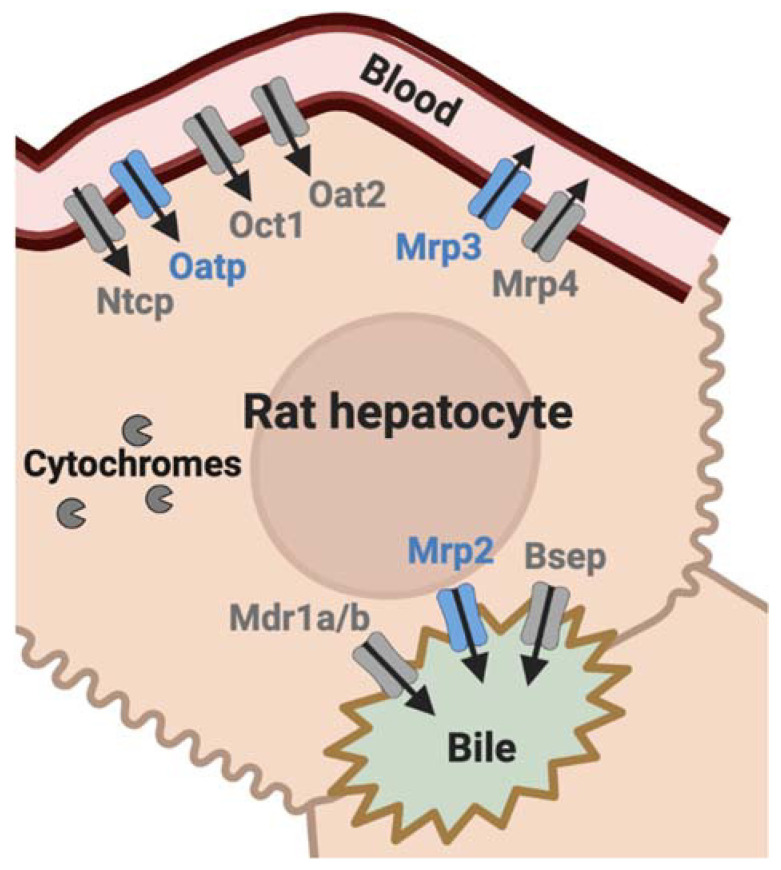
Membrane transporters expressed in rat hepatocytes. Transporters known to be involved in the hepatobiliary transport of ^99m^Tc-mebrofenin are highlighted in blue. Bsep: bile salt export pump; Mdr: multidrug resistance; Mrp: multidrug resistance-associated protein; Ntcp: Na+-taurocholate cotransporting polypeptide; Oat: organic anion transporter; Oatp: organic anion-transporting polypeptide; Oct: organic cation transporter.

**Figure 2 pharmaceuticals-15-00392-f002:**
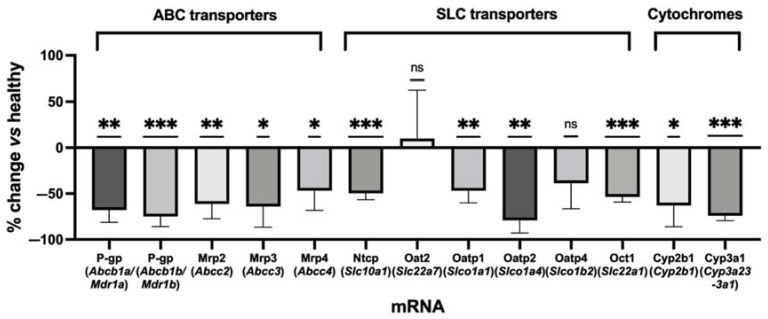
Changes in transporter and cytochrome mRNA expression in the liver of LPS-treated rats (24 h after exposure to LPS) compared to healthy rats estimated by qPCR (*n* = 4 both in healthy and LPS-treated groups). Data are mean ± SD. * *p* ≤ 0.05, ** *p* ≤ 0.01, *** *p* ≤ 0.001, ns indicates not significant, one-sample *t*-test after a Shapiro–Wilk Normality test. Cyp: cytochrome; LPS: lipopolysaccharide; Mdr: multidrug resistance; Mrp: multidrug-resistance associated protein; Ntcp: Na+-taurocholate cotransporting polypeptide; Oat: organic anion transporter; Oatp: organic anion-transporting polypeptide; Oct: organic cation transporter.

**Figure 3 pharmaceuticals-15-00392-f003:**
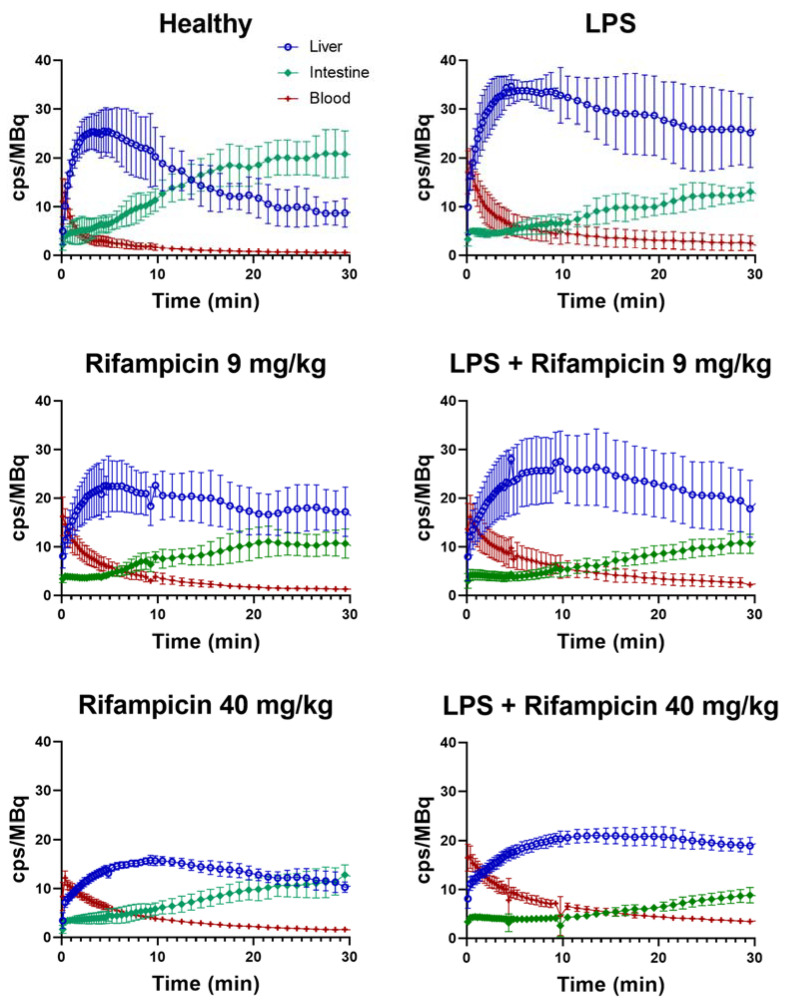
Mean (±SD) time–activity curves of ^99m^Tc-mebrofenin in the liver (○), intestine (◆), and blood (+) of healthy and LPS-treated rats (24 h after exposure to LPS) under baseline conditions and after treatment with 9 or 40 mg/kg rifampicin (*n* = 5 for control animals and *n* = 6 for LPS-treated animals). Radioactivity is expressed as counts per second (cps) normalized to the injected dose (MBq). LPS: lipopolysaccharide. Data obtained in the healthy and Rifampicin 40 mg/kg groups have already been presented in a previous study [29].

**Figure 4 pharmaceuticals-15-00392-f004:**
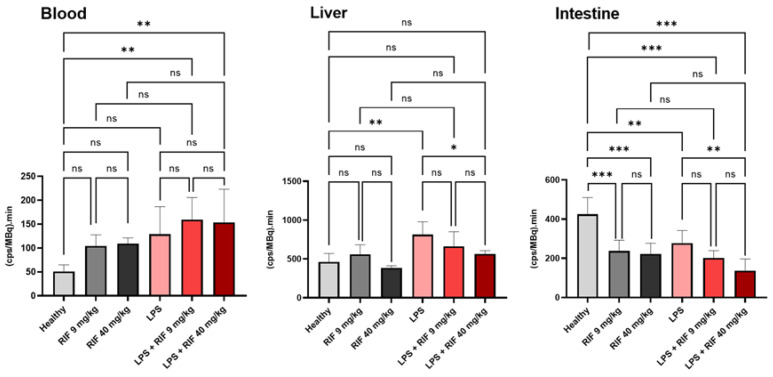
Area under the time–activity curves (AUC) of ^99m^Tc-mebrofenin in the blood pool (imaged-derived), liver, and intestine obtained in healthy (*n* = 5 per group) and in lipopolysaccharide (LPS)-treated rats (*n* = 6 per group). Impact of pre-treatment with 9 or 40 mg/kg of rifampicin (RIF), a potent Oatp/Mrp2/Mrp3 inhibitor is reported. Radioactivity is expressed in counts per second (cps) in each region of interest, normalized to the injected radioactivity amount in each animal (cps/MBq). Data are mean ± SD. * *p* < 0.05, ** *p* < 0.01, *** *p* < 0.001, ns indicates not significant, ordinary one-way ANOVA followed by a Tukey’s post hoc test for multiple comparison. LPS: lipopolysaccharide; RIF: rifampicin. Data obtained in the healthy and RIF 40 mg/kg groups have already been presented in a previous study [29].

**Figure 5 pharmaceuticals-15-00392-f005:**
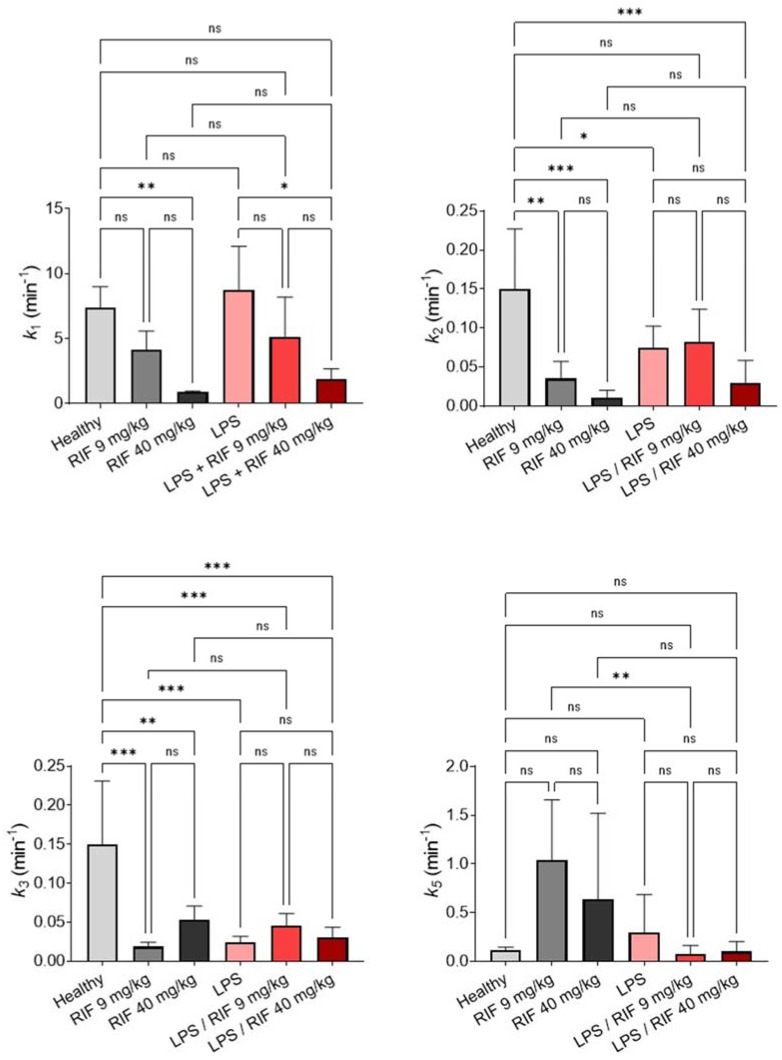
Pharmacokinetic model outcome parameters describing the hepatobiliary transport of ^99m^Tc-mebrofenin in healthy and LPS-treated rats (24 h after exposure to LPS) under baseline conditions and after treatment with 9 or 40 mg/kg rifampicin (*n* = 5 for control animals and *n* = 6 for LPS-treated animals). Data are mean ± SD. * *p* ≤ 0.05, ** *p* ≤ 0.01, *** *p* ≤ 0.001, ns not significant, ordinary one-way ANOVA followed by a Tukey’s post hoc test for *k*_2_ and *k*_3_, and Kruskal–Wallis test for comparison of *k*_1_ and *k*_3_ data. LPS: lipopolysaccharide; RIF: rifampicin. The data of healthy and RIF 40 mg/kg groups have already been presented in a previous study [29].

## Data Availability

Data is contained within the article and Appendix A.

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
