# Peer review of "Pharmacokinetic Imaging Using 99mTc-Mebrofenin to Untangle the Pattern of Hepatocyte Transporter Disruptions Induced by Endotoxemia in Rats"

_pharmaceuticals, 2022, doi:10.3390/ph15040392_

Round 1

Reviewer 1 Report

The work of Marie et al. describes the application of 99mTc-mebrofenin to evaluate the impact of endotoxemia (LPS administration) and DDIs (rifampicin administration) on ABC and SLC liver transporters and cytochromes. The conclusions are supported by experimental data and the paper complements the work already presented by the authors for this radiotracer in previous publications.

The presented information is generally new, but I have a few issues:

Comments:

Figure 4: similarly to what has been done in Figure 2/Table S1 and Figure 5/Table S2, please include exact AUC values in a supplementary table.

Lines 162-163: “(…) the parameter precision (assessed by percentage coefficient of variation, %CV) was acceptable (Table S2).” Please specify the % CV values considered acceptable, since there are CV values of 69.9 or 125.7%.

In Table S2 the authors state that pharmacokinetic parameters are presented. Nevertheless, only transfer rate constants are shown, instead of PK parameters such as clearance, AUC or half-life. Therefore, I suggest altering the name of the table.

Furthermore, some of the values presented in this table, particularly for healthy rats and rifampicin 40 mg/kg, have already been published by the authors in Table S1 of Marie et al. Pharmaceutics. 2020;12(6):486. Although the purpose of these papers is different, I advise a reference to this previous publication in the table headline.

Figure S1 – Have the time-activity curves of 99mTc-mebrofenin in the liver and intestine for healthy rats and rifampicin 40 mg/kg been published elsewhere? If so, please include references to previous papers.

Figure S2 – This figure has already been presented by the same authors in Marie et al. Pharmaceutics. 2020;12(6):486. I suggest its removal from the present publication or inclusion with a citation of the first paper and a different figure caption, in order to avoid self-plagiarism.

Author Response

We thank the reviewer for his/her constructive comments. We have addressed all points he/she has raised and changed the manuscript accordingly.

Figure 4: similarly to what has been done in Figure 2/Table S1 and Figure 5/Table S2, please include exact AUC values in a supplementary table.

As suggested by the reviewer, AUC values relative to Fig. 4 have been added in Table S2 (supplementary materials and lines 386 - 388).

We noticed that AUC values presented in Fig. 4 in the initial manuscript were not corrected by the injected dose of 99mTc-mebrofenin. Although the injected dose of 99mTc-mebrofenin was relatively homogeneous across groups (37.4 ± 5.0 MBq, i.v.), we think it is important to report dose-normalized data, which is consistent with our previous work using the same method. Please note that significance is lost when comparing dose-normalized “blood” data of some groups using ANOVA and post-hoc analysis (blood, Figure 4). This has no impact on the results of PK modeling and did not change the conclusions of the manuscript. Please see the corrected results lines 149-155.

Lines 162-163: “(…) the parameter precision (assessed by percentage coefficient of variation, %CV) was acceptable (Table S2).” Please specify the % CV values considered acceptable, since there are CV values of 69.9 or 125.7%.

This is now clarified lines 195– 199 and lines 222-224.

In Table S2 the authors state that pharmacokinetic parameters are presented. Nevertheless, only transfer rate constants are shown, instead of PK parameters such as clearance, AUC or half-life. Therefore, I suggest altering the name of the table.

The title of Table S2 (now Table S3) has been changed accordingly (supplementary materials and line 390).

Furthermore, some of the values presented in this table, particularly for healthy rats and rifampicin 40 mg/kg, have already been published by the authors in Table S1 of Marie et al. Pharmaceutics. 2020;12(6):486. Although the purpose of these papers is different, I advise a reference to this previous publication in the table headline.

Thank you for suggesting. The reference to the previously published data used for comparison with the new groups is now clearly mentioned lines 147-148, 183-185, 231-232, Table S2, Table S3 and Figure S1.

Figure S1 – Have the time-activity curves of 99mTc-mebrofenin in the liver and intestine for healthy rats and rifampicin 40 mg/kg been published elsewhere? If so, please include references to previous papers.

This is now clearly mentioned, please see our previous answer.

Figure S2 – This figure has already been presented by the same authors in Marie et al. Pharmaceutics. 2020;12(6):486. I suggest its removal from the present publication or inclusion with a citation of the first paper and a different figure caption, in order to avoid self-plagiarism.

We followed the reviewer’s advice and removed Figure S2. The reference to the previous paper remains and the reader will be able to access the figure in the other (open access) paper.

Reviewer 2 Report

Dear colleagues, 

The article has an important topic, but some little changes must be done

  1. the abstract section is hard to understand, please rewrite to be more clearly
  2. in the introduction section, put a phrase regarding the role of hepatobiliary scintigraphy in the diagnostic of hepatobiliary disorders 

Author Response

We thank the reviewer for his/her constructive comments. We have addressed all points he/she has raised and changed the manuscript accordingly.

  1. the abstract section is hard to understand, please rewrite to be more clearly

We have now structured the abstract for improved clarity (lines 19-35)

  1. in the introduction section, put a phrase regarding the role of hepatobiliary scintigraphy in the diagnostic of hepatobiliary disorders 

A sentence has been added about this topic (lines 91-98)

Other changes to the initial manuscript:

We noticed that AUC values presented in Fig. 4 in the initial manuscript were not corrected by the injected dose of 99mTc-mebrofenin. Although the injected dose of 99mTc-mebrofenin was relatively homogeneous across groups (37.4 ± 5.0 MBq, i.v.), we think it is important to report dose-normalized data, which is consistent with our previous work using the same method. Please note that significance is lost when comparing dose-normalized “blood” data of some groups using ANOVA and post-hoc analysis (blood, Figure 4). This has no impact on the results of PK modeling and did not change the conclusions of the manuscript. Please see the corrected results lines 149-155.